# Genetic Variation and Stability Analysis of an Artificially Synthesized Allohexaploid *Brassica* for Breeding Innovations

Su Yang [1,2,3], Kangni Zhang [2], Chenze Lu [1], Guangna Chen [1], Qian Huang [2], Zaid Ulhassan [2], Ji'an Wei [3], Muhammad Ahsan Farooq [2] and Weijun Zhou [2,*]

1   Key Laboratory of Specialty Agri-Products Quality and Hazard Controlling Technology of Zhejiang Province, College of Life Sciences, China Jiliang University, Hangzhou 310018, China
2   Institute of Crop Science and Zhejiang Key Laboratory of Crop Germplasm, Zhejiang University, Hangzhou 310058, China
3   Mizuda Group Co., Ltd., No. 288, Tianziwei Road, Huzhou 313000, China
*   Correspondence: wjzhou@zju.edu.cn; Tel.: +86-571-8898-2770

**Abstract:** Allopolyploids play an essential role in plant evolution and confer apparent advantages on crop growth and breeding compared to low ploidy levels. A doubled haploid (DH) population derived from the cross between two artificially synthesized allohexaploid *Brassica* was created and self-crossed continuously. Morphological and yield-related traits showed considerable variation among different generations, different families and even within the same families. However, the flowering time, pollen viability and seed yield increased gradually during the selfing process. Ploidy level estimation and karyotyping analysis revealed that this population was chimeras with varied chromosome numbers within an identical plant. Chromosome translocations analysis showed that the B genome was more instable compared to the A and C genomes. The A genome was more prone to chromosome recombination than the C genome. Although some genomic regions were more likely to be duplicated, deleted, or rearranged, a consensus pattern was not shared between different progenies. This research deepened our understanding of the genetic variation of artificially synthesized allohexaploid *Brassica*. In addition, the allohexaploid *Brassica* can be used as a bridge to transfer some of the valuable traits blocked by reproductive barriers from wild *Brassica* species to cultivated species such as cold and drought resistance, etc.

**Keywords:** allohexaploid *Brassica*; polyploid breeding; phenotypic traits; genetic variation

## 1. Introduction

*Brassica* species include some of the most important oil crops (canola and rapeseed), vegetables (cabbage, turnip, broccoli, cauliflower, pak choi and bok choy) and condiments (mustard and pickles) in the world and are the most important agricultural species in the *Brassicaceae* family [1]. *Brassica* species are rich in dietary fiber, vitamin C, phytosterols and contain beneficial anti-carcinogenic compounds. Moreover, using *Brassica* species as renewable raw materials has attracted growing interest in the biofuel and chemical industries [2]. *Brassica* oilseeds are the second largest oil crop in the world; their output is second only to soybean. Among them, the yield and planting area of *B. napus* was well ahead of *B. rapa* and *B. juncea*. However, the germplasm resources of *B. napus* are relatively narrow due to its short taming time and traditional breeding methods. With the emergence of global warming, various extreme weather events put forward new requirements on *B. napus* about cold, drought, disease, and insect resistance. For this, there is the utmost need to expand the germplasm resources of *Brassica* oilseeds and produce new cultivars/species with better quality (oil and fiber) and higher seed yield to replace the existing cultivars [3].

As described in the Triangle of U [4], *Brassica* comprises three basic diploid species including *B. rapa* (AA, 2n = 20), *B. nigra* (BB, 2n = 16), *B. oleracea* (CC, 2n = 18) and three amphidiploids including *B. napus* (AACC, 2n = 38), *B. juncea* (AABB, 2n = 36) and *B. carinata*

(BBCC, $2n$ = 34) [5]. Each of the above amphidiploids evolved from a hybridization event between a pair of the three diploids. For example, *B. napus* is obtained by the natural hybridization and chromosome doubling between *B. rapa* and *B. oleracea*. All the species mentioned above have useful agronomic traits and carry different disease resistance characteristics [6–13]. Combing the three genomes from U's triangle can generate new haplotypes/hybrids with better agronomic traits.

Interspecific hybridization and polyploidization, the processes by which new allopolyploid species are created, are known to confer significant advantages: hybrid vigor, increased environmental tolerances and transgressive segregation for parental species traits have all been identified in allopolyploids relative to their lower ploidy progenitors [14]. Additionally, since there are marked differences among the A, B and C genomes [15], interspecific hybridization among the *Brassica* crops could broaden the genetic pool by combining the A, B and C genomes. Large numbers of hybridization and polyploidy events have been found before the formation of polyploid crops. Approximately 70% of angiosperms have experienced one or more chromosome doubling events during their evolutionary history [16]. Low frequencies of naturally occurring or spontaneous interspecific hybridization events have also been detected between the six *Brassica* "U's Triangle" species [17,18]. However, no naturally occurring species with all three genomes (AABBCC, $2n$ = 54; allohexaploid) exists.

Combining the three genomes of six species in U's triangle to create a novel allohexaploid *Brassica* is the long-term goal of *Brassica* breeding [2,19]. Many breeders have attempted to synthesize trigenomic hexaploid *Brassica* in the last few decades. In 1942, Howard [20] reported the first synthesized allohexaploid *Brassica*. Later, Iwasa [21] attempted to synthesize fertile allohexaploid with stable meiosis. Currently, there are three strategies for synthesizing allohexaploid. The first and most important one is to cross the amphidiploids with the corresponding diploid species, which is followed by chromosome doubling to obtain allohexaploid *Brassica*. The second method is to cross the diploid species twice among three diploid species accompanied by two chromosome doubling events to obtain allohexaploid *Brassica*. For example, cross *B. rapa* (AA) with *B. oleracea* (CC) followed by colchicine treatment to obtain hybrid (AACC), then cross it with *B. nigra* (BB) followed by another round of colchicine treatment to obtain allohexaploid *Brassica*. Another method is to cross twice between three different amphidiploids and take advantage of unreduced gametes rather than somatic doubling to increase the ploidy level [22–24]. For example, cross *B. juncea* (AABB) with *B. carinata* (BBCC) to obtain hybrid (ABBC), then use the unreduced gametes to cross with *B. napus* (AACC) to obtain allohexaploid *Brassica*.

Even though hybridization and chromosome doubling bring about competitive advantages to allopolyploids, they also have adverse effects. Usually, synthetic polyploids will undergo meiotic synapsis disorder [25]. Newly synthesized allohexaploids will experience rapid changes in genome composition and gene expression including chromosome recombination, loss of chromosome fragments, gene activation and silencing, changes in the expression level of homologous genes and even chromosome deletion, etc. [26–29]. Many chromosome rearrangements have been observed among different chromosomes of the same genome and among different genome components in the polyploid genome by genetic mapping and genome in situ hybridization [5,30]. Genetically, polyploids will reduce the gene expression ability to the level of diploid ancestors to avoid gene redundancy [31]. Genomic instability is a major obstacle in the process of polyploid formation [25], which would significantly affect the successful construction of the allohexaploid *Brassica* as well. Although many attempts have been made to synthesize allohexaploid *Brassica* through different approaches, an utterly stable allohexaploid *Brassica* remains elusive due to chromosome instability and poor seed fertility [19]. To date, only a few studies have examined the morphological variation, fertility, and meiotic stability over subsequent generations of hexaploid *Brassica* [3,32,33]. More studies regarding the stability and fertility of allohexaploid *Brassica* are necessary for establishing a fertile and stable allohexaploid *Brassica* species.

Different allohexaploid *Brassica* synthesis approaches were used to create a novel allohexaploid hybrid H16-1 with high allelic diversity, and 235 DH lines were created by microspore culture in our previous research [34]. After studying the stability and phenotype of these DH lines, 189 DH lines with stable chromosome numbers ($2n = 54$) and wide genetic variation were selected to form a DH population [30]. A high-density genetic linkage map was constructed based on SSR and SNP markers followed by the QTL mapping of yield-related traits [5,35]. Since these allohexaploid *Brassica* materials were artificially synthesized by hybridization and chromosome doubling, many abnormal cells were produced during meiosis in $F_1$ generation, such as the formation of lagging chromosomes, triads, polyads and cells with unequal division, etc. [34]. In this research, the DH progenies from the allohexaploid hybrid *Brassica* H16-1 were self-crossed for multiple generations, and the morphological and yield-related traits, ploidy level, chromosome composition, and genetic variation were studied. It is expected that by this study, we could have a deeper understanding of the stability of artificially synthesized allopolyploid materials and find new allohexaploid *Brassica* materials with good traits and stable inheritance.

## 2. Materials and Methods

### 2.1. Plant Material

The maternal parent 7H170-1 was synthesized through interspecific crosses between *B. rapa* and *B. carinata* [3], while the paternal parent Y54-2 was generated from the cross between *B. napus* and *B. nigra* [36]. Both the parents of hybrid H16-1 and the hybrid itself were confirmed to have a complete set of hexaploid complement ($2n = 54$) by chromosome checking [34]. A DH population (here, we regarded it as $S_0$ generation) of hybrid H16-1 was established through microspore culture [34]. A total of 442 siblings derived from 42 lines from $S_0$ generation were sown under the filed condition at the University of Western Australia ($S_1$ generation). After harvesting, these plants were divided into three groups: C1, C2 and C3. When the seed yields per plant of all progenies from the same line are all higher than 2 g, these progenies will be placed in the C1 group (high-yield progenies from high-yield families). If the seed yield of all progenies from the same line are less than 2 g, these progenies will be placed in the C3 group (low-yield progenies from low-yield families). If both high-yield and low-yield progenies exist in the same line, these low-yield progenies be placed in the C2 group (low-yield progenies from high-yield families). Thirty-five lines from these three categories were randomly chosen to grow at the greenhouse at the University of Western Australia with three replicates each (Table 1). Based on the performance of $S_2$ generation, 20 lines were randomly chosen to grow at the glasshouse of Zhejiang University with ten replicates each ($S_3$ generation).

**Table 1.** The classification, germination rate and ploidy level estimation of $S_2$ generation derived from an allohexaploid *Brassica* hybrid H16-1.

| $S_1$ Generation | Seed Yield (g/Plant) | Selection Category | $S_2$ Generation | Germination Rate (%) | Mean Hexaploid Percentage (%) | Standard Deviation | Coefficient of Variation (%) |
|---|---|---|---|---|---|---|---|
| 7-6-088 | 2.216 | C1 | 6-1 | 92.86 | 15.98 | 0.03 | 18.78 |
| 7-6-088 | 2.485 | C1 | 6-2 | 92.86 | 18.27 | 0.04 | 21.89 |
| 7-6-203 | 3.485 | C1 | 6-3 | 92.86 | 26.53 | 0.15 | 55.40 |
| 7-6-053 | 2.273 | C1 | 6-4 | 78.57 | 19.73 | 0.06 | 27.88 |
| 7-6-107 | 2.109 | C1 | 6-5 | 71.43 | 35.75 | 0.09 | 24.05 |
| 7-6-107 | 6.355 | C1 | 6-6 | 100.00 | 59.09 | 0.13 | 21.32 |
| 7-6-204 | 5.076 | C1 | 6-7 | 42.86 | 42.56 | 0.26 | 59.92 |
| 7-6-204 | 3.091 | C1 | 6-8 | 78.57 | 47.45 | 0.23 | 48.68 |
| 7-6-279 | 5.663 | C1 | 6-9 | 64.29 | 34.99 | 0.13 | 36.87 |
| 7-6-050 | 3.063 | C1 | 6-10 | 92.86 | 40.79 | 0.15 | 35.55 |
| 7-6-050 | 5.993 | C1 | 6-11 | 71.43 | 50.31 | 0.13 | 25.84 |

**Table 1.** *Cont.*

| $S_1$ Generation | Seed Yield (g/Plant) | Selection Category | $S_2$ Generation | Germination Rate (%) | Mean Hexaploid Percentage (%) | Standard Deviation | Coefficient of Variation (%) |
|---|---|---|---|---|---|---|---|
| 7-6-323 | 14.802 | C1 | 6-12 | 100.00 | 48.61 | 0.05 | 11.11 |
| 7-2-003 | 2.452 | C1 | 6-13 | 78.57 | 37.73 | 0.14 | 37.64 |
| 7-6-103 | 2.349 | C1 | 6-14 | 50.00 | 40.41 | 0.07 | 16.08 |
| 7-6-267 | 3.684 | C1 | 6-15 | 60.00 | 27.63 | 0.15 | 54.28 |
| 7-6-267 | 4.855 | C1 | 6-16 | 64.29 | 22.71 | 0.09 | 37.87 |
| 7-6-267 | 2.126 | C1 | 6-17 | 71.43 | 22.22 | 0.03 | 13.95 |
| 7-6-30 | 6.031 | C1 | 6-18 | 64.29 | 41.91 | 0.20 | 46.76 |
| 7-6-30 | 4.956 | C1 | 6-19 | 85.71 | 36.91 | 0.19 | 52.56 |
| 7-6-30 | 4.849 | C1 | 6-20 | 100.00 | 29.43 | 0.11 | 38.39 |
| 7-6-30 | 3.985 | C1 | 6-21 | 100.00 | 25.89 | 0.08 | 31.29 |
| 7-6-30 | 2.975 | C1 | 6-22 | 57.14 | 30.45 | 0.06 | 20.36 |
| 7-6-30 | 33.965 | C1 | 6-23 | 92.86 | 13.70 | 0.02 | 13.14 |
| 7-6-30 | 0.171 | C2 | 6-24 | 50.00 | 37.51 | 0.20 | 54.12 |
| 7-6-053 | 0.07 | C2 | 6-25 | 78.57 | 38.83 | 0.09 | 23.18 |
| 7-6-204 | 0.162 | C2 | 6-26 | 85.71 | 21.51 | 0.03 | 13.48 |
| 7-6-050 | 0.109 | C2 | 6-27 | 85.71 | 27.03 | 0.05 | 18.50 |
| 7-2-003 | 0.104 | C2 | 6-28 | 78.57 | 41.26 | 0.15 | 35.14 |
| 7-6-267 | 0.331 | C2 | 6-29 | 78.57 | 25.87 | 0.14 | 52.57 |
| 7-4-001 | 0.056 | C3 | 6-30 | 85.71 | 52.04 | 0.15 | 29.21 |
| 7-4-001 | 0.111 | C3 | 6-31 | 85.71 | 51.82 | 0.19 | 36.66 |
| 7-4-001 | 0.039 | C3 | 6-32 | 100.00 | 55.55 | 0.17 | 31.14 |
| 7-4-001 | 0.043 | C3 | 6-33 | 50.00 | 52.36 | 0.03 | 5.92 |
| 7-6-080 | 0.094 | C3 | 6-34 | 64.29 | 53.31 | 0.19 | 35.27 |
| 7-6-080 | 0.027 | C3 | 6-35 | 85.71 | 51.23 | 0.01 | 2.73 |

*2.2. Phenotype Characterization*

Seeds were germinated in plug trays with moist substrates in the growth chamber at 22/18 °C (day/night) with a 16 h photoperiod at a light intensity of 300 µmol m$^{-2}$ s$^{-1}$ and relative humidity at 60%. After nine days, the plug trays were shifted to vernalization room for one month. Later, the vernalized seedlings were transplanted into flowerpots in the glass house. Morphological traits including the branch number, leaf thickness, leaf color, leaf margin, leaf shape, flower and pod shape of these allohexaploid *Brassica* materials were observed. The flowering time was recorded at the flowering stage. In addition, plant height, seed yield, thousand seed weight (TSW) and above-ground biomass were measured at the harvesting stage.

*2.3. Flow Cytometry Analysis*

The ploidy level was measured following Geng et al. [34] with a few modifications. Approximately 5 mg young leaf samples were collected in lysis buffer (15 mM Tris-HCl, 80 mM KCl, 20 mM NaCl, 20 mM EDTA-Na$_2$, 15 mM mercaptoethanol, 0.05% TritonX-100), chopped by a razor, filtered with 300 mesh screens, incubated with 2 µL RNase (3 mg/mL) for 30 min and stained with a 50 ng/mL PI solution. A BD FACSCanto II (BD Biosciences, San Jose, CA, USA) flow cytometer with a 488 nm laser was used to analyze the stained nuclei samples. The single tube mode was used to collect more than 20,000 cells per sample, and FlowJo V7.2.5 (FlowJo Software, USA) was used to analyze the data. The DNA content of samples was estimated based on the mean value of G1 peak: sample 2C DNA content = ((sample G1 peak mean)/(standard G1 peak mean) × [standard 2C DNA content (pg) and the reference standard (*B. napus*) with a known genome DNA content of 2.29 pg.

*2.4. Karyotyping by Multi-Color Fluorescence In Situ Hybridization (FISH)*

The multi-color FISH experiment was completed following Kato et al. [37] and Xu et al. [38]. Tender flower buds (1–2 mm) were collected with tweezers and stored

in Carnoy's solution. Constantly replace the solution until it is no longer discolored. Rinse the flower buds with distilled water 2–3 times and treat with a mixture of 20% cellulase and 20% pectinase for 2 h at 37 °C followed by 75 mM KCl for 45 min. Add one drop of glacial acetic acid and use tweezers to disperse the flower bud cells. Examine the slides under the phase contrast microscope and choose mitotic metaphase slides for multi-color FISH. 5S rDNA, 26S rDNA, KBrH092N24, CentBrI, CentBrII, pBrSTR and pBNBH35 probes were prepared by nick translation. Add 200 μL of deionized formamide to the prepared slides and denature them at 80 °C for 10 min. Then, the slides were dehydrated by ethanol gradient (70%, 90%, 100%) and dried in the air. Add 30–40 μL hybrid solution (50% deionized formamide, 10% dextran sulfate, 10 μL labeled probe, 0.5 μL denatured salmon sperm DNA and 5 μL 20 × SSC) onto the slides, cover and seal them. Denature the slides at 80 °C for 10 min and hybridize them overnight at 37 °C in a moisturizing box. After that, rinse the slides under 2 × SSC and 0.4 × SSC twice at 40 °C, respectively. Add 10 μL DAPI staining solution after the slides dried, cover the slides, and observe the florescent signal intensity under a ZEISS Imager M2 fluorescence microscope (Carl Zeiss, Oberkochen, Germany), and digital images were captured using the AxioCamMRm software (Carl Zeiss, Oberkochen, Germany). Images were cropped, size-adjusted, and contrast-optimized using only functions affecting the whole image with Adobe Photoshop.

### 2.5. Pollen Viability Estimation

Pollen viability was measured by staining the pollen grains from three newly opened flowers in beautiful weather with 1% acetocarmine. At least 300 pollen grains in each sample were checked under microscope. Big, round, and stained pollen grains are fertile, while small, flat, and unstained ones are infertile [39].

### 2.6. Chromosome Translocations Analysis

The approximate centromere locations for the A and C genome chromosomes were established according to the *B. napus* Darmor-bzh v8.1 reference genome [40] based on the remapping of previous RAD-seq data [5]. Here, 500 Kb was set as an interval to observe the frequency of insertion and deletion events. The darker the color, the higher the frequency of insertion or deletion events in this interval. Missing regions covering less than 1 Mb were not considered in the analysis due to consideration of Indel distribution and density constraints. The translocations between the genomes were established based on homoeologous relationships between A, B and C genomes [40]. The final karyotype was plotted using the R package RIdeogram [41].

### 2.7. Statistical Analysis

The SPSS 20.0 (SPSS, Chicago, IL, USA) statistical package was used to analyze the data. One-way variance analysis (ANOVA) was carried out, which was followed by Duncan's multiple range test ($p < 0.05$) to establish significant differences for the morphological and physiological parameters among different families, groups, and generation with three biological replicates each. The coefficient of variance (CV) of different families was studied to analyze the stability and variation situation in each family. When the CV value is lower than 15%, it is considered stable. Correlation analysis was performed using Pearson correlation analysis. Canonical variate analysis was completed using Genestat (2016, 18th edition, VSN International Ltd, Hemel Hempstead, UK).

## 3. Results

### 3.1. Morphological Variation

This population was derived from the cross between two artificially synthesized allohexaploid *Brassica*. Although the chromosome number of parents and $F_1$ hybrid was 54, their genome structure was quite unstable. The drastic changes in the genome caused large numbers of phenotypic variation and brought about many new features. In $S_2$ generation, plants showed different morphological traits including plant height, branching

type, flowering time, leaf, pod shape, etc. At the same growth period, some plants showed good vegetative growth with delayed bolting; some grew weak with late bolting, while some grew weak but already bolted (Figure 1a–c). Some plants bloomed but were sterile and produced empty pods, while others were fertile with many seeds in pods (Figure 1d,e). Some plants have limited branches, while others have many (Figure 1f,g). Some plants are highly tall (Figure 1h). The morphology of leaves, flowers and pods also varied greatly among different $S_2$ progenies (Figure 2). Some plants had thick, curling and deep-colored leaves, while others had thin and light-colored leaves (Figure 2a,b). There were round and oval leaves, leaves with two or four pairs of stipules, curly leaves, or leaves with purple petioles (Figure 2c–h). For flowers, there were three-petaled flowers, over-opened petals, petals with anthers shorter than stigmas, petals with similar length and width, crinkled petals, abnormal petals, etc. (Figure 2i–o). There was also a significant variation in the span of siliques, pedicels, beaks, and width of the pods. Some pods were plump with long siliques, pedicels, and medium beaks; some were shriveled with long siliques, short pedicels, and long beaks; some had average pods with medium siliques, long pedicels, and medium beaks, while some pods were empty with short siliques, medium pedicels, short beaks, etc. (Figure 2p–s). Some individuals showed similar phenotypes to that of their diploid and tetraploid ancestors (*B. napus* and *B. rapa*) in this generation.

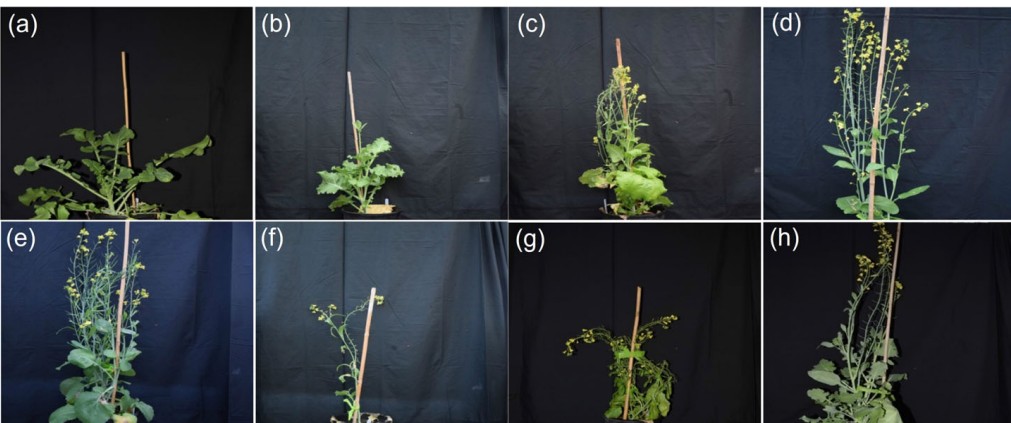

**Figure 1.** Phenotypic variation in $S_2$ generation derived from a hexaploid *Brassica* hybrid H16-1. (**a**) Good vegetative growth with delayed bolting. (**b**) Weak growth with late bolting. (**c**) Weak growth but bolted. (**d**) Blossomed with low fertility (empty pods). (**e**) Blossomed with high fertility (pods with many seeds inside). (**f**) Few branches. (**g**) Many branches. (**h**) Tall plant.

### 3.2. Ploidy Level Estimation

Flow cytometry was used to evaluate the DNA content of the allohexaploid *Brassica* materials in $S_2$ generation and found that almost all plants were chimeras with varied chromosome numbers within an identical plant. Generally, there were four different types of chimeras: large numbers of hexaploid cells and a small number of triploid cells (Figure 3c), lots of triploid cells and a few hexaploid cells (Figure 3d), equal numbers of triploid and hexaploid cells (Figure 3e), cells containing diploids, tetraploids, and a small number of hexaploids (Figure 3f). Most progenies belonged to the first three types. Only one progeny (6-1-1) belonged to the fourth type. Among the first three types, the second type had the highest frequency.

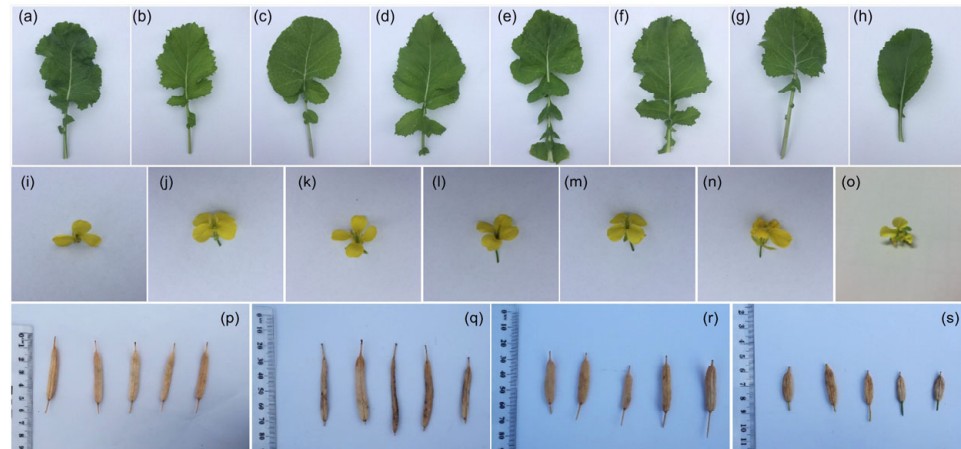

**Figure 2.** Morphological variation of leaves, flowers and pods in S$_2$ generation derived from an allohexaploid *Brassica* hybrid H16-1 DH population. (**a**) Thick, curled and deep-colored leaves. (**b**) Thin leaves with light color. (**c**) Round leaves. (**d**) Two pairs of stipules. (**e**) Four pairs of stipules. (**f**) Curl leaves. (**g**) Purple petioles. (**h**) Oval leaves. (**i**) Flower with three petals. (**j**) Normal flowers. (**k**) Over-opened petals. (**l**) Anthers shorter than stigma. (**m**) Petals with similar length and width. (**n**) Crinkled petals. (**o**) Abnormal flowers. (**p**) Plump pods with long siliques, long pedicels and medium beaks. (**q**) Shriveled pods with long siliques, short pedicels and long beaks. (**r**) Average pods with medium siliques, long pedicels, and medium beaks. (**s**) Empty pods with short siliques, medium pedicels and short beaks.

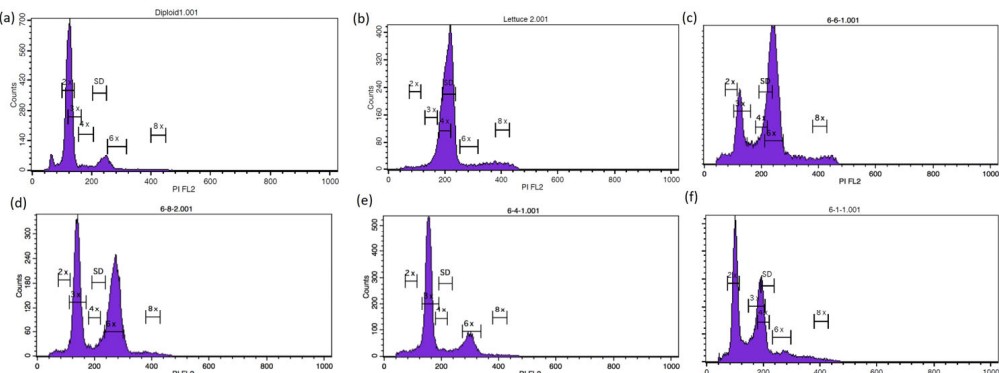

**Figure 3.** Examples of ploidy determination by flow cytometry in S$_2$ generation derived from an allohexaploid *Brassica* hybrid H16-1. (**a**) Diploid control. (**b**) Lettuce control. (**c**) Large numbers of hexaploid cells and a small number of triploid cells. (**d**) Lots of triploids and a few hexaploids. (**e**) Equal numbers of triploid and hexaploid cells. (**f**) Cells containing diploids, tetraploids, and a small number of hexaploids.

The average proportion of hexaploid cells in the C1, C2 and C3 groups in S$_2$ generation was 33.44%, 32.00% and 52.72%, respectively (Table 1). The ploidy level of the C3 group was significantly higher than those in the C1 and C2 groups (Table 2). There was no plant with 100% hexaploid cells. 6-6-1 has the highest hexaploid percentage (72.88%), while 6-23-2 has the lowest hexaploid rate (11.96%). The average CV in S$_2$ generation was 45.20%, showing large variation. Among the 35 families, six families were stable (6-12, 6-17, 6-23, 6-26, 6-33 and 6-35) while the remaining 29 families were relatively unstable. 6-35 had the lowest CV (2.81%), whereas 6-7 had the highest CV (59.97%).

**Table 2.** Phenotypic variation among three different groups (C1, C2 and C3 goup) in $S_2$ generation derived from a hexaploid *Brassica* hybrid H16-1.

| Phenotypic Traits | C1 Group | C2 Group | C3 Group |
|---|---|---|---|
| Hexaploid percentage (%) | 33.44% ± 0.16 [b] | 32.00% ± 0.13 [b] | 52.72% ± 0.12 [a] |
| Pollen viability (%) | 37.59% ± 0.32 [a] | 48.26% ± 0.31 [a] | 46.50% ± 0.22 [a] |
| Plant height (cm) | 145.62 ± 35.28 [a] | 153.28 ± 33.12 [a] | 118.06 ± 33.69 [b] |
| Flowering time (day) | 86.61 ± 14.60 [a] | 92.56 ± 16.37 [a] | 83.33 ± 7.24 [a] |
| Above-ground biomass (g) | 48.38 ± 21.36 [a] | 55.64 ± 24.53 [a] | 19.39 ± 11.98 [b] |
| Seed yield (g) | 2.61 ± 2.86 [a] | 1.33 ± 1.35 [ab] | 0.31 ± 0.42 [b] |
| 1000-seed weight (g) | 4.66 ± 0.90 [a] | 4.50 ± 1.68 [a] | 4.73 ± 2.04 [a] |

Data are the means of at least three replicates ± standard error (S.E). Values followed by the different letters indicate significant differences based on one-way ANOVA followed by Duncan's test ($p < 0.05$).

### 3.3. Karyotype Analysis

To further confirm the results of ploidy level estimation, multi-color FISH was carried out to precisely locate each chromosome. When observing the mitotic metaphase of $S_2$ generation under the microscope, we noticed that the chromosome number of progenies from the same families varied a lot. Moreover, chromosome numbers in the same plant were also different, further indicating that materials in this generation were chimeras with unstable chromosome composition. After accurately identifying each chromosome in $S_2$ generation, we found that the B genome was very unstable in this population. Large numbers of chromosomes deletion happened in the B genome. For example, in 6-10-1 (41 chromosomes), every chromosome in the A and C genomes existed, while only three chromosomes in the B genome remained and the remaining 13 chromosomes were missing (Figure 4). Both chromosome deletion and addition occurred in 6-22-3 (39 chromosomes). In this plant, the A and C genome were had an extra C3 chromosome while the B genome only had two B2 chromosomes left (Figure 4). This kind of situation also happened in 6-13-1 (52 chromosomes). Chromosome loss happened in the A1, C3 and C4 chromosomes while chromosome addition happened in the C1 chromosome (Figure 4). However, we also found that in different cells of 6-13-1, the observed chromosome numbers were different. In the four cells we observed in this material, the chromosome numbers were 52, 35, 43 and 53. We also noticed that two cells have the complete set of B genome chromosomes in 6-13-1, one cell lost the B genome, and the other cell only had five B genome chromosomes. 6-29-2 (44 chromosomes) had the complete A genome but lost one C3 chromosome and 9 B genome chromosomes. This result was consistent with the results of ploidy level observation, which indicated that there were chimeras in this population.

### 3.4. Pollen Vitality Evaluation

The pollen viability of $S_2$ and $S_3$ generation was observed. $S_2$ generation showed large variation (0 to 95.94%) with an average of 40.95% (Figure 5). Nine progenies were totally infertile, while six progenies had pollen viability of higher than 90%. The proportion of low fertility (≤30%), medium fertility (30–60%) and high fertility (≥60%) in $S_2$ generation were 40.00%, 30.48% and 29.52%, respectively (Figure 6). Most of the plants showed a downward tendency in pollen viability compared with their diploid or tetraploid ancestors. Among the 35 families, 6-18 and 6-21 were almost infertile with the fertility of every plant less than 2%. Seven out of the 35 (20%) families were relatively stable (CV < 15%). The remaining 28 families were somewhat unstable and highly variable. In the C1, C2 and C3 groups, the average pollen viability was 37.59%, 48.26% and 46.50%. There was no significant correlation between groups and pollen viability (Table 2), suggesting that higher seed yield plants did not necessarily produce higher pollen viability progenies. In $S_3$ generation, the average pollen viability increased to 55.76%, ranging from 0 to 92.22%. Only two plants in $S_3$ generation were totally infertile. The proportion of low, medium, and high fertility plants in $S_3$ generation were 13.24%, 38.97% and 52.85% (Figure 6). Obviously, the pollen viability in $S_3$ generation has increased significantly compared to $S_2$ generation. When

referred to the stability of different families in S$_3$ generation, six out of 20 (30%) families were stable, indicating that S$_2$ and S$_3$ families showed similar variation.

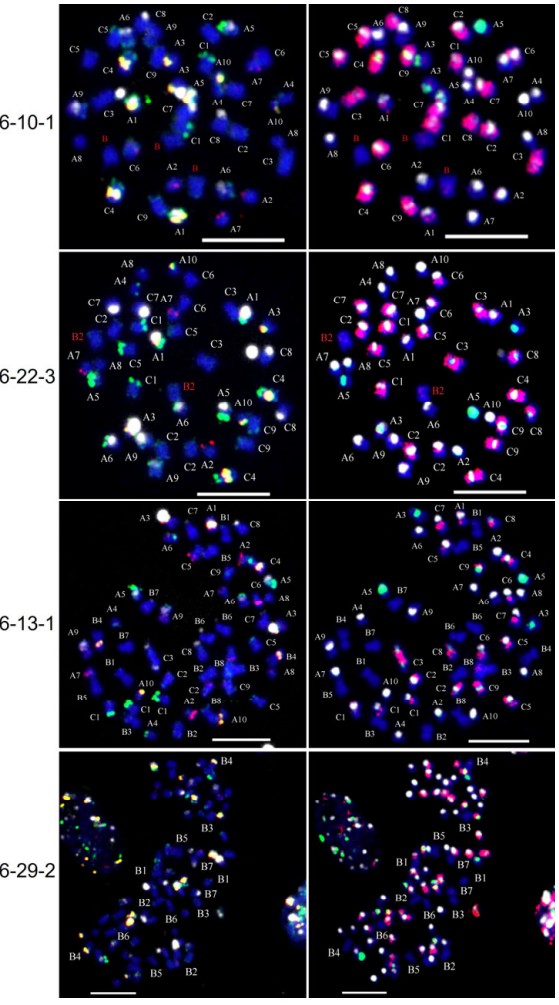

**Figure 4.** The karyotype analysis in S$_2$ generation derived from an allohexaploid *Brassica* hybrid H16-1 by multi-color FISH. The signal of 5S rDNA, 26S rDNA, pBrSTR and KBr092N24 showed orange, white, green and red color, respectively. The chromosomes were stained blue by DAPI. All scale bars were 10 μm.

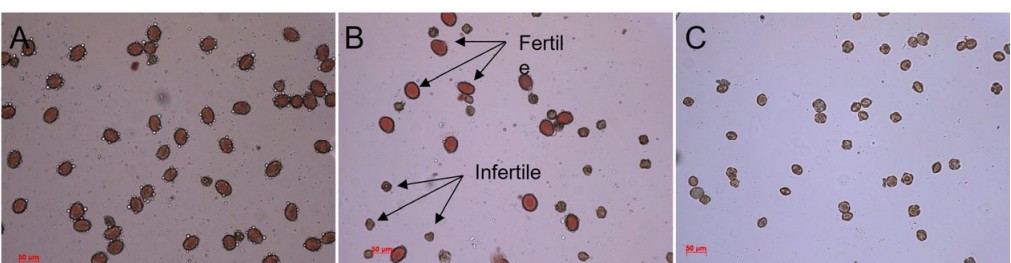

**Figure 5.** Pollen viability in S$_2$ generation derived from an allohexaploid *Brassica* hybrid H16-1 DH population. (**A**) High fertility. (**B**) Medium fertility. (**C**) Low fertility.

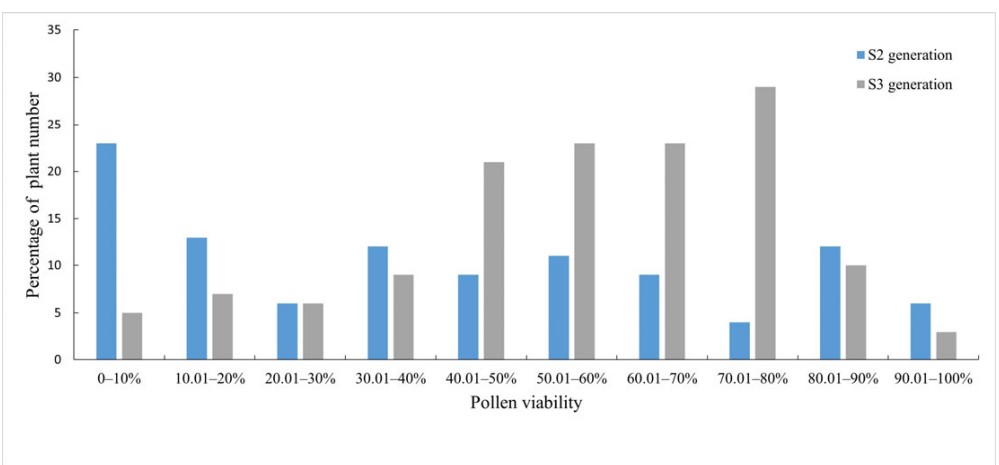

**Figure 6.** Statistical result of pollen viability in $S_2$ and $S_3$ generation derived from an allohexaploid *Brassica* hybrid H16-1.

*3.5. Agronomic Traits Analysis*

Flowering time is one of the most critical traits in the plant developmental process. In $S_1$ generation, the average flowering time in 32 families (144 progenies) was 120.10 days ranging from 76 to 177 days. In the 35 families (105 progenies) in $S_2$ generation, the flowering time varied from 65 to 130 days with an average of 87.07 days. The mean flowering time of C1, C2 and C3 groups were 86.61, 92.56 and 83.33 days with the CV of 16.86%, 17.68% and 8.68%, respectively. The percentage of stable families in $S_1$ and $S_2$ generations were 43.33% and 60.00%, indicating that the stability was increasing gradually from generation to generation.

The plant height of $S_2$ generation varied from 63 to 235 cm, with an average height of 142.20 cm (Table 3). Overall, 54.29% of the families were relatively stable. The average plant height of 6-8 was 201.67 cm. C1, C2 and C3 groups had an average height of 145.62, 153.28 and 118.06 cm. The plant height of both C1 and C2 groups were significantly higher than that of the C3 group (Table 2). These results indicated that higher seed yield plants in $S_2$ generation tended to have progenies with higher plant height. The average CV in these three groups were 13.90%, 11.89% and 26.26%. Compared to the other two groups, the C3 group had a shorter plant height and displayed a larger variation.

**Table 3.** The correlation analysis among different traits in $S_2$ generation derived from an allohexaploid *Brassica* hybrid H16-1.

| | Flowering Time | Chromosome Number | Pollen Viability | Seed Yield | 1000-Seed Weight (TSW) | Plant Height | Above-Ground Biomass |
|---|---|---|---|---|---|---|---|
| Flowering time | 1 | | | | | | |
| Chromosome number | 0.290 ** | 1 | | | | | |
| Pollen viability | 0.036 | −0.211 * | 1 | | | | |
| Seed yield | −0.076 | −0.228 * | 0.368 ** | 1 | | | |
| 1000-seed yield | 0.233 * | 0.085 | 0.019 | −0.137 | 1 | | |
| Plant height | 0.210 * | 0.023 | −0.076 | 0.264 ** | 0.044 | 1 | |
| Above-ground biomass | 0.007 | −0.125 | 0.201 * | 0.453 ** | −0.079 | 0.653 ** | 1 |

Pearson's chi-squared test ($p < 0.05$) was used to test the level of significance. * and ** represented the level of significant at 0.05 and 0.01 probability levels, respectively.

The above-ground biomass varied from 2.95 to 110.72 g, with an average biomass of 44.66 g. 6-27-3 had the highest above-ground biomass, while 6-35-3 was the lowest. Nearly one-third (31.43%) of all the families in $S_2$ generation were stable (CV < 15%). The above-ground biomass of every plant in family 32 was lower than 20 g, while every plant in family 10 was higher than 80 g. The average above-ground biomass in C1, C2 and

C3 groups were 48.38, 55.64 and 19.39 g. The above-ground biomass of both the C1 and C2 groups were significantly higher than that of the C3 group ($p < 0.05$) (Table 2). It can be inferred that higher seed yield plants in $S_2$ generation tended to have progenies with higher above-ground biomass. The CVs of these three groups were 25.50%, 26.73% and 61.27%. Families in the C3 group were thinner and weaker than those in the C1 and C2 groups with larger variation. This was in accordance with the plant height in different families in $S_2$ generation.

Among the 105 plants characterized in $S_2$ generation, seven plants harvested no seeds at all. The average seed yields and TSW in the remaining 98 plants in $S_2$ generation were 2.12 and 4.64 g (Table S1). Seed yield ranged from 0 to 13.15 g and TSW ranged from 0.82 to 8.00 g in $S_2$ generation (Table S1). Seed yield was very diversified in $S_2$ generation, while TSW was relatively stable compared to seed yield. The average seed yield in the C1, C2 and C3 groups was 2.61, 1.33 and 0.31 g. The average TSW in these three groups was 4.66, 4.50 and 4.73 g. The seed yield of the C1 group was significantly higher than that in the C3 group, while no significant differences was found in TSW among the three groups ($p < 0.05$) (Table 2). It can be inferred that higher seed yield in $S_1$ generation tended to have progenies with higher seed yield.

Correlation analysis was conducted among different traits in $S_2$ generation (Table 3). Results showed that seed yield was significantly positively correlated with pollen viability, plant height and above-ground biomass. Among them, the correlation coefficient between seed yield and above-ground biomass was the highest. This indicated that by improving pollen viability, plant height or above-ground biomass, seed yield could be significantly enhanced. However, we noticed that seed yield was negatively correlated with chromosome number. This might be due to the instability of hexaploidy *Brassica* materials. Plants with more chromosomes tended to be more unstable, resulting in a low seed yield. Apart from seed yield, the chromosome number was significantly negatively correlated with pollen viability but significantly positively correlated with flowering time. In addition, the flowering time was significantly correlated with TSW and plant height. Above-ground biomass was significantly correlated with pollen viability and plant height.

Canonical variates analysis was performed among the three groups (C1, C2 and C3) in $S_2$ generation regarding seven phenotypic traits (hexaploid percentage, seed yield, 1000-seed weight, plant height, above ground biomass, pollen viability and flowering time). Only two canonical variates were obtained, among which CV1 explained 78.73% of the population stability and CV2 explained 21.27% (Table 4). The corresponding equation is 5.2188 hexaploid percentage + 0.0102 plant height + 0.0074 seed yield—0.1704 1000-seed weight—0.0406 above-ground biomass—0.0328 flowering time—0.0555 pollen viability. A scatter plot was drawn with CV1 and CV2 (Figure 7). Results showed that the C1 group had some overlapping with the C2 and C3 groups, respectively. However, there is no overlapping between the C2 and C3 groups, showing huge differences.

**Table 4.** The characteristic value, contribution rate and overall contribution rate of canonical variate analysis of three groups (C1, C2 and C3) in $S_2$ generation derived from an allohexaploid *Brassica* hybrid H16-1.

| Canonical Variate | Characteristic Value | Contribution Rate (%) | Overall Contribution Rate (%) |
|---|---|---|---|
| CV1 | 0.4471 | 78.73 | 78.73 |
| CV2 | 0.1208 | 21.27 | 100 |

### 3.6. Fixed Chromosome Translocations

Indels were used to analyze the presence of fixed translocations between A and C genomes in $S_2$ generation (Figure 8). After comparing the indels derived from RAD-seq with the *B. napus* reference genome, it was found that large numbers of insertion and deletion events occurred in the A and C genomes, but the frequency was inconsistent. Generally, the frequency of indels (both insertion and deletion) in the A genome was significantly higher than that in the C genome, indicating that the A genome was more

unstable than the C genome. To be more specific, the frequency of insertion and deletion events in 10 chromosomes in the A genome were relatively high; meanwhile in the C genome, C1, C3, C6, C7 and C8 showed a high frequency of insertion and deletion events, and the other four chromosomes (C2, C4, C5, C9) showed a relatively low percentage of indels events. For most chromosomes, the frequency of insertion and deletion regions was consistent. For example, there was a dense region of insertion and deletion in the upper part of chromosome A3 and A9. However, we also noticed that the frequency of insertion and deletion regions was unevenly distributed in some regions, such as chromosome A7 and C5.

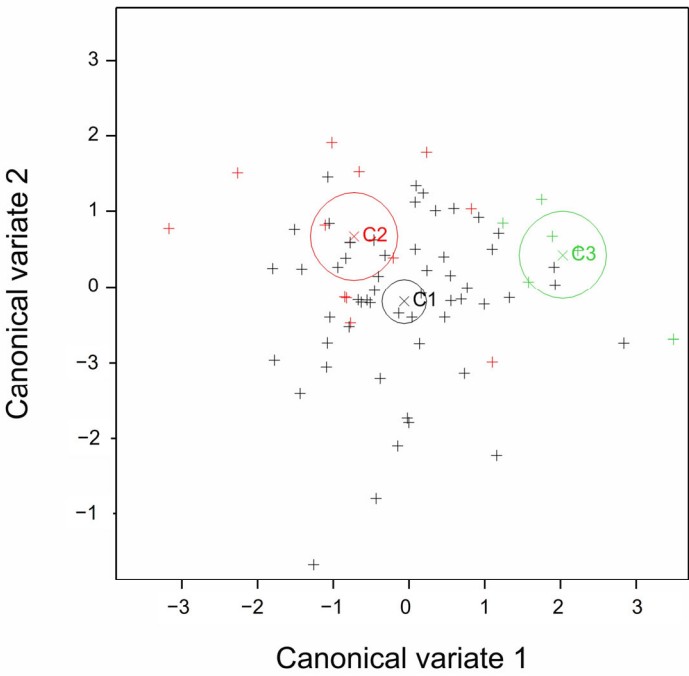

**Figure 7.** The canonical variate analysis of three groups (C1, C2 and C3) in $S_2$ generation derived from an allohexaploid *Brassica* hybrid H16-1.

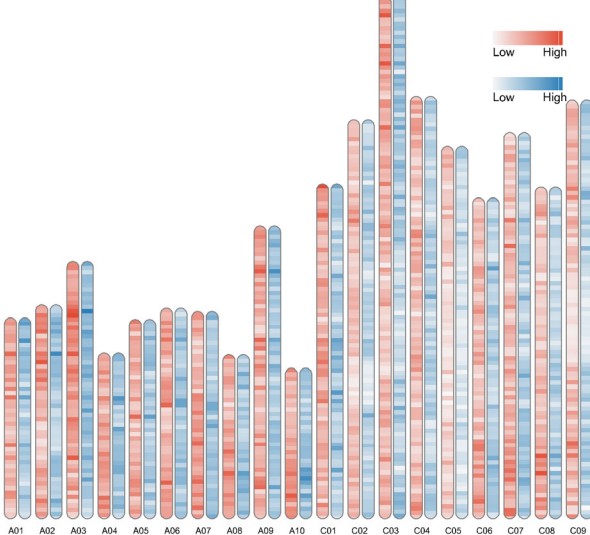

**Figure 8.** Insertion and deletion events in A and C genomes in $S_2$ generation derived from an allohexaploid *Brassica* hybrid H16-1. Red color represents insertion events and bule color represents deletion events. The darker the color, the higher the frequency, and vice versa.

## 4. Discussion

Polyploidization is a vital force to promote the formation of new species and evolution, but the newly formed polyploid genome used to be unstable and needed to experience a violent oscillation process. During this process, some special phenotypes will appear, such as death, autogenic transformation, etc. Gene structure will also change, including chromosome recombination, inversion and translocation, sequence loss, etc. [42,43]. Polyploid wheat and cotton, which widely exist in nature, have experienced chromosome instability and even chromosome loss in the early stage. However, after several generations, the meiotic pairing of chromosomes has gradually become normal and stable [44,45]. However, for synthetic new polyploids that do not exist in nature, chromosome instability and chromosome loss are more serious. Therefore, it needs more generations of selection to obtain genetically stable polyploid materials. Since there are only three diploid *Brassica* species and three tetraploid species in the U's triangle, no naturally allohexaploid *Brassica* exists. So, the artificially synthesized allohexaploid *Brassica* and their progenies are extremely chromosomal unstable, showing lots of phenotypic and genetic variation.

Flow cytometry analysis showed that most $S_2$ progenies were composed of hexaploid and triploid cells, while the minority consisted of tetraploid and diploid cells, indicating that materials in this generation were chimeras. Our hexaploid materials were obtained by a wide hybridization between *B. carinata* and *B. rapa*, *B. napus* with *B. nigra*, which was followed by chromosome doubling. However, these artificially synthesized hexaploid materials were not stable due to meiotic synaptic disorder. During the selfing process, some cells remained hexaploidy, some cells returned to the state before chromosome doubling and became triploid cells, while others lost one to several chromosomes. 6-1-1 had chromosome numbers similar to their diploid and tetraploid parents, indicating that this material was extremely unstable and tended to return to the status before wide hybridization.

Karyotyping results showed that not only chromosome number of progenies from the same families varied a lot, but also chromosome numbers in the same plant were inconsistent, proving that plants in this generation are chimeras with unstable chromosome composition. This is in accordance with the ploidy level detection results; both showed that the chromosome in this generation is quite unstable due to meiosis disorder. Allohexaploid *Brassica* must overcome the major challenge of establishing regular meiosis to become a new polyploid [46]. Irregular chromosome pairing is caused by the three ancestrally homologous chromosome sets from different evolutionary lineages, which are known as homologous chromosomes [47]. If nonhomologous pairing during meiosis occurs, it can lead to different chromosomal rearrangements, such as deletions, duplications, and translocations [48], heavily affecting genome stability. It is also noticed that the B genome is more prone to chromosome loss. This is possibly caused by the high homology between the A and C genomes, which can occur as a result of normal pairing during meiosis, while the B genome is disordered during synapsis, resulting in uneven chromosome distribution. As a result, in the same material, some cells have a complete B genome set, and some cells have completely lost the B genome or only a small number of B genome chromosomes. Quezada Martinez et al. [49] also found that more potential new rearrangements happened in the B genome of the $F_1$ hybrids. Once the B genome is lost, the A and C genomes remained in the material, thus becoming a new allotetraploid *Brassica* material. This new material showed similar phenotypic traits to that of *B. napus*. However, since these materials underwent complex chromosomal recombination events, they differed from their tetraploid parent. If these new tetraploid materials can be stably inherited, novel *B. napus* can be generated from them, thus expanding the germplasm resources of *B. napus*.

Due to meiosis disorder, allohexaploid *Brassica* materials may experience chromosome recombination, inversion and translocation, sequence loss, and the chromosome number is unstable or even missing [49]. Indel analysis showed that lots of chromosome rearrangements have happened in $S_0$ generation, and the A genome is more prone to chromosome recombination events than the C genome. Although some genomic regions were found to

be more likely to be duplicated, deleted, or rearranged, a consensus pattern was not shared between genotypes.

The parents of our allohexaploid *Brassica* H16-1 were derived from interspecific hybridization between *B. carinata* and *B. rapa*, *B. napus* and *B. nigra*. The pollen viability in these diploid and tetraploid parents were all above 90% [35,50]. However, the average pollen viability in $S_2$ generation was only 40.95%, ranging from 0 to 95.94%. The reduction in pollen fertility of artificially synthesized polyploids had also been reported in other research. Sundberg et al. [51] discovered that the pollen fertility of resynthesized *B. napus* decreased from 38% to 70%. The fertility of autotetraploid Maize can reduce by 85% to 95% while autotetraploid *Gossypium herbaceum* is almost sterile [52]. No usual pairing between homologous chromosomes and unusual pairing between non-homologous chromosomes lead to a failure of meiosis, resulting in lots of abnormal phenomena such as chromosome lagging or early separation and reduced pollen fertility [3]. On the other hand, newly synthesized allohexaploids will experience rapid changes in genome composition and gene expression [26–29]. Differences in the chromosome structure and spontaneous chromosome breakage and reunion can cause abnormalities related to abnormal pollen production, thus leading to poor pollen fertility [53]. In $S_3$ generation, the average pollen fertility increased to 55.76% with a range of 0 to 92.22%. Unlike the generally consistent distribution of pollen fertility in $S_2$ generation, the proportion of low, medium, and high fertility increased gradually. This suggests that fertility is recovering progressively from $S_2$ to $S_3$ generation. Since the fertility of synthesized hexaploid is usually unstable, it is believed that by continuously selfing, the chromosome pairing could be ameliorated, thus obtaining progenies with increased pollen fertility [3].

$S_2$ generation showed plenty of morphological variation including plant growth, leaf morphology, flower morphology, pod size and shape. Tian et al. [3] obtained hexaploid materials by crossing *B. carinata* and *B. rapa* followed by chromosome doubling. They also found significant morphological differences between parents and hexaploid progenies as well as among different progenies. The variation among different individuals of $S_2$ generation in this experiment also provides the possibility for the direct utilization of some unique traits such as drought and disease resistance.

Comparing three categories of the $S_2$ progenies derived from the $S_1$ lines with different yield performances, the average yield of C1 showed a downtrend from 5.60 to 2.54 g/plant, but it was still higher than that of C2 (increased from 0.16 to 1.33 g/plant) and C3 (increased from 0.06 to 0.28 g/plant). When combined with the results of ploidy identification, the mean hexaploid percentage of the C3 group was significantly higher than that of C1 and C2. It revealed that the ploidy level had a negative association with seed yield, which might be due to the high frequency of homoeologous exchanges in allopolyploids. Canonical variates analysis showed that all the phenotypic traits can be divided into two CV1s and CV1 can explain as large as 78.73% of the population stability. From the equation of CV1, we can find that hexaploid percentage has a large effect on the stability of this population. This result is in accordance with the ploidy level comparison. It has been reported that meiotic stability was the primary factor influencing fertility across several generations of self-pollinated allohexaploid plants [54]; therefore, selection for high fertility is also expected to realize the breeding objective of increased meiotic stability [55].

Seed yield was significantly correlated with pollen fertility, plant height and above-ground biomass. By improving pollen fertility through continuous selfing and choosing progenies with higher pollen fertility, we could obtain allohexaploid *Brassica* with high seed yield. However, both seed yield and pollen fertility were negatively correlated with chromosome number in $S_2$ generation. This may be attributed to the fact that allohexaploid *Brassica* tended to become stable after chromosome deletion. However, the specific reasons need to be further explored.

CV analysis showed that higher seed yield plants in $S_1$ generation tended to have progenies with higher seed yield, plant height and above-ground biomass and lower hexaploid percentage. However, they do not necessarily lead to differences in pollen

viability, TSW and flowering time in $S_2$ generation. It hints to us that by selecting plants with higher seed yield in the last generation, we could obtain progenies with higher seed yield, taller plants, and larger above-ground biomass. Among the 35 lines, 6, 7, 19, 11 and 1 lines were stable in consideration of hexaploid percentage, pollen viability, plant height, above-ground biomass, and seed yield, respectively. Among the 35 families, 6-3, 6-4, 6-10, 6-21 and 6-26 were relatively stable. Interestingly, all $S_2$ progenies, even from the same line, appeared to have a considerable variation in the trait of seed yield, which might be attributed to the fact that seed yield is determined by multiple yield-related traits, such as silique number per plant, seed number per silique, and TSW, and it is also easily affected by the environment [56]. In conclusion, almost half of the detected $S_2$ lines were relatively stable, and we suggested that 7-6-053-4 was the most stable line based on comprehensive indexes.

## 5. Conclusions

In this research, complicated interspecific hybridization among several important *Brassica* species and stability analysis has been carried out to establish a novel allohexaploid *Brassica* species. Results showed that this population showed sizeable morphological variation among different generations, different families from the same generation and even within same families. Nevertheless, it is noticed that the flowering time, pollen viability, and seed yield increased gradually during the selfing process. Genetic analysis revealed that the progenies were all chimeras and the B genome was more prone to chromosome loss. The A genome was more likely to undergo chromosome recombination compared to the C genome. Although none of the $S_2$ progenies had the expected 2n = 54 chromosome karyotype as a stable allohexaploid, it is feasible to improve stability over generations. For instance, after five generations of selfing with the selection of the DH population, we have selected three promising lines as breeding materials due to their potential stability, good agronomic characteristics, and high fertility rate [19]. Once a stable allohexaploid *Brassica* is established, it could provide opportunities to analyze interactions between three genomes of *Brassica* species and to investigate the evolution of *Brassica*. Moreover, valuable traits such as drought and disease resistance from four cultivated species, *B. rapa*, *B. nigra*, *B. carinata* and *B. napus*, are combined in the hexaploid itself, and new transgressive traits might be created and selected by breeding. In addition, the particular origins and genome composition of the hexaploid *Brassica* enable it to be used as a bridge to transfer genes or traits to *Brassica* crops such as *B. napus*, *B. juncea* and *B. carinata*, which were usually blocked by interspecific reproductive barriers.

**Supplementary Materials:** The following supporting information can be downloaded at: https://www.mdpi.com/article/10.3390/agronomy12112843/s1, Table S1: Yield-related traits of $S_2$ generation derived from an allohexaploid *Brassica* hybrid H16-1.

**Author Contributions:** W.Z. and S.Y. conceived the idea, S.Y., K.Z., Z.U. and M.A.F. designed and wrote the manuscript, C.L., G.C., Q.H. and J.W. revised the manuscript. All authors have read and agreed to the published version of the manuscript.

**Funding:** This research was supported by the Natural Science Foundation of Zhejiang Province (LQ20C130006), the Science and Technology Department of Zhejiang Province (2021C02064-2), Collaborative Innovation Center for Modern Crop Production co-sponsored by Province and Ministry (CIC-MCP), and the Agriculture and Rural Affairs Department of Zhejiang Province (2021XTTGLY0202).

**Data Availability Statement:** Not applicable.

**Acknowledgments:** Part of the experiment was conducted at the University of Western Australia under the supervision of Wallace A. Cowling, Sheng Chen and Guijun Yan. The authors sincerely thank them.

**Conflicts of Interest:** The authors declare no conflict of interest.

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
