# Peer review of "Genetic Variation and Stability Analysis of an Artificially Synthesized Allohexaploid Brassica for Breeding Innovations"

_agronomy, doi:10.3390/agronomy12112843_

Round 1

Reviewer 1 Report

Manuscript "Genetic Variation and Stability Analysis of an Artificially Synthesized Allohexaploid Brassica for Breeding Innovations" is very interesting.

General comments:

Authors analyzed the DH progenies from the allohexaploid hybrid Brassica H16-1. DH lines were self-crossed for multiple generations. Authors studied the morphological and yield-related traits, ploidy level, chromosome composition, and genetic variation.

Detailed comments:

Palnt material is very good, but unfortunately description of "2.7. Statistical Analysis" is very poor. Lack is information about distribution of observed traits. Lack is information about correlation analysis (for maens?).
Table 1 needs homogeneous groups.
"Figure 6. Statistical result of pollen": Statistical? What method?

My suggestion:

Authors should apply canonical variate analysis for comparison of DH lines for all traits jointly.
Genetic parameters? Estimation of additive and epistatic effects?

Paper needs major revision.

Author Response

Dear editors and reviewers:

Thank you for your decision and constructive comments on our manuscript entitled “Genetic Variation and Stability Analysis of an Artificially Synthesized Allohexaploid Brassica for Breeding Innovations” (Manuscript ID: agronomy-1976030). Those comments were very valuable and helpful for revising and improving our manuscript. We have carefully considered the suggestion of reviewers and tried our best to improve and modify the manuscript accordingly. Revised portion can be found in the manuscript with track change. The main corrections in the manuscript and the responds to the reviewers’ comments are as following:

General comments:
Authors analyzed the DH progenies from the allohexaploid hybrid Brassica H16-1. DH lines were self-crossed for multiple generations. Authors studied the morphological and yield-related traits, ploidy level, chromosome composition, and genetic variation.

Detailed comments:
Palnt material is very good, but unfortunately description of "2.7. Statistical Analysis" is very poor. Lack is information about distribution of observed traits. Lack is information about correlation analysis (for maens?).

Response: Thank you for your valuable feedback. We have added the detailed description of “2.7”. We observed the phenotypic traits including hexaploid percentage, pollen vitality, plant height, flowering time, above-ground biomass, seed yield and 1000-seed weight from S1 to S3 generation. One way ANOVA was carried out followed by Duncan’s multiple range test (P<0.05) to establish the significant differences for these phenotypic traits among three different groups (C1, C2 and C3) in S2 generation and for the flowering time and pollen viability among three generation (S1, S2 and S3) with three biological replicates each. The coefficient of variance (CV) of different families was studied to analyze the stability and variation situation in each family. When the CV value is lower than 15%, it is considered stable. Correlation analysis was performed using Pearson correlation analysis.

Table 1 needs homogeneous groups.

Response: We appreciate the reviewer for this kind recommendation. Since the instability and variation of our allohexaploid Brassica, plants from the same line were regarded as different progenies to explore the genetic stability of these materials. So, we divided the 35 families in S2 generation into three group (C1, C2 or C3). If both high-yield (higher than 2 g) and low-yield (lower than 2 g) progenies exist in the same line, these high-yield progenies go to C1 group (high yield progenies from high yield families) and those low-yield progenies go to C2 group (low yield progenies from high yield families). Therefore, Table 1 was ordered by selection group. In this case, some progenies from the same line might be divided into different groups. For example, 7-6-30 had 6 progenies (6-18 to 6-24), 5 of which (6-18 to 6-23) belonged to C1, 6-24 belonged to C2 group. Similar situation occurred in7-6-053, 7-6-204, 7-6-050, 7-2-003 and 7-6-267. Therefore, in Table 1, the same lines seemed to appear in different places and were not classified according to homogeneity. But in fact, although these lines were from the same parents, they are already different from each other genetically, so we regarded them as separated lines and classified them by selection group.

"Figure 6. Statistical result of pollen": Statistical? What method?
Response: Thank you for your question. We want to show the distribution of pollen viability of different individuals in S2 generation. So, we have corrected it as follows: Distribution of pollen activity in S2 and S3 generations derived from an allohexaploid Brassica hybrid H16-1.

My suggestion:
Authors should apply canonical variate analysis for comparison of DH lines for all traits jointly.
Genetic parameters? Estimation of additive and epistatic effects?
Response: Thank you very much for your precious suggestion. We have performed canonical variate analysis using GenStat Windows 18th Edition (VSN International 2016). The results, table and figure can be found in the manuscript. Please kindly check them.

For genetic parameters, we have estimated the ploidy level by flow cytometry, checked the karyotype by multi-color FISH. Besides, we did further Indel analysis based on previous RAD-seq data (Yang et al. 2018) to explore the chromosome translocation events in this population. You have mentioned the estimation of additive and epistatic effects in this population. We have already done this analysis with SNPs discovered RAD-seq in our previous paper (Yang et al. 2018). We detected 25 additive QTL controlling six phenotypic traits including seed number, seed yield, pod length, plant height, 1000-seed weight, pollen viability, and 62 epistatic QTL pairs controlling four phenotypic traits (seed number, seed yield, 1000-seed weight, and pollen viability). Several interesting candidate genes for these traits were discovered at two QTL hotspots. So, we re-analyzed the sequencing data to analyze the insertion and deletion events happened in this population in this manuscript.

Reviewer 2 Report

It is an interesting work about the genetic variation of artificially synthesized allohexaploid Brassica. The authors created a doubled haploid (DH) population derived from the cross between two artificially synthesized allohexaploid Brassica and self-crossed continuously. They performed different genetic variations and stability analyses.

However, the manuscript has some minor concerns before acceptance of the article for publication.

Line 129-131: By what criteria was this division done? How much yield was considered high or low yield? It should be added.

Line 131: How were they selected? What were the selection criteria?

Line 194-196: The experimental design information including the type and number of replication should be added.

Results: I did not find the results of the means comparisons using Duncan in the text as a Table or Chart (with ranks a, b, c, ....)!!

Author Response

Dear editors and reviewers:

Thank you for your decision and constructive comments on our manuscript entitled “Genetic Variation and Stability Analysis of an Artificially Synthesized Allohexaploid Brassica for Breeding Innovations” (Manuscript ID: agronomy-1976030). Those comments were very valuable and helpful for revising and improving our manuscript. We have carefully considered the suggestion of reviewers and tried our best to improve and modify the manuscript accordingly. Revised portion can be found in the manuscript with track change. The main corrections in the manuscript and the responds to the reviewers’ comments are as following:

Comments and Suggestions for Authors

It is an interesting work about the genetic variation of artificially synthesized allohexaploid Brassica. The authors created a doubled haploid (DH) population derived from the cross between two artificially synthesized allohexaploid Brassica and self-crossed continuously. They performed different genetic variations and stability analyses.

However, the manuscript has some minor concerns before acceptance of the article for publication.

Line 129-131: By what criteria was this division done? How much yield was considered high or low yield? It should be added.

Response: Thank you for your question. It's true that we didn't make it clear here. After harvesting S1 generation, we found that the yield of different progenies from the same DH line varied a lot. To facilitate the research, we divided the 42 lines (442 progenies) in S1 generation into three groups, and randomly choose 35 progenies from these three groups as S2 generation. More specifically, when the seed yield per plant is higher than 2 g, it is considered as high yield; when it is lower than 2 g, it is regarded as low yield. When the seed yield of all progenies from the same line are all higher than 2 g, these progenies will be divided into C1 group; If the seed yield of all progenies from the same line are less than 2 g, these progenies will be divided into C3 group; If both high-yield and low-yield progenies exist in the same line, these low-yield progenies go to C2 group and those high-yield progenies go to C1 group.

Line 131: How were they selected? What were the selection criteria?

Response: Thanks for your valuable comments. We have added the detail. After divided the 42 lines (442 progenies) in S1 generation into three groups, we randomly choose 35 progenies from these three groups as S2 generation for further research. If we grow all the 442 progenies with three replicates each, there will be too many experimental materials. The difficulty of the experiment will increase, and the accuracy of the experiment will be difficult to guarantee. Therefore, we randomly select several progenies as representatives for further observation. In this way, we can understand why the seed yield of some groups are always high, some groups are always low, and some groups varies greatly.

Line 194-196: The experimental design information including the type and number of replications should be added.

Response: Thank you for your suggestion. SPSS 20.0 (SPSS, Chicago, IL, USA) statistical package was used to analyze the data. One-way variance analysis (ANOVA) was carried out, followed by Duncan’s multiple range test (P<0.05) to establish significant differences for the morphological and physiological parameters (ploidy level, pollen viability, flowering time, plant height, seed yield, TSW, above-ground biomass, et al.) among different families, groups, and generation with three biological replicates each.

Results: I did not find the results of the means comparisons using Duncan in the text as a Table or Chart (with ranks a, b, c, ....)!!

Response: Thank you for this valuable feedback. Previously we have done Duncan’s multiple range test (P<0.05) for various phenotypic traits among different families, groups, and generation, but we didn’t show the results in tables or charts cause we thought it might be difficult to present such a large amount of data. So, we just presented the results in the plain text. After further consideration, we made a new table showing the mean value comparison among three groups in S2 generation (Table 2).

Round 2

Reviewer 1 Report

Now, all is ok.